# Psychological Stress-Induced Pathogenesis of Alopecia Areata: Autoimmune and Apoptotic Pathways

**DOI:** 10.3390/ijms241411711

**Published:** 2023-07-20

**Authors:** Dongkyun Ahn, Hyungjun Kim, Bombi Lee, Dae-Hyun Hahm

**Affiliations:** 1Department of Medicine, College of Medicine, Kyung Hee University, Seoul 02447, Republic of Korea; dannytty@khu.ac.kr; 2KM Science Research Division, Korea Institute of Oriental Medicine, Daejeon 34054, Republic of Korea; heyjoon73@kiom.re.kr; 3Center for Converging Humanities, Kyung Hee University, Seoul 02447, Republic of Korea; 4Department of Physiology, College of Medicine, Kyung Hee University, Seoul 02447, Republic of Korea

**Keywords:** alopecia areata, stress, substance P, corticotropin-releasing hormone, immune privilege

## Abstract

Alopecia areata (AA) is an autoimmune dermatological disease with multifactorial etiology and is characterized by reversible hair loss in patches. AA may be closely related to emotional stress and influenced by psychological factors as part of its pathophysiology; however, its etiology remains predominantly unknown. This review aimed to elucidate the association between AA occurrence and the neuropeptide substance P (SP) and corticotropin-releasing hormone (CRH), which are secreted during emotional stress, and have been understood to initiate and advance the etiopathogenesis of AA. Therefore, this review aimed to explain how SP and CRH initiate and contribute to the etiopathogenesis of AA. To assess the etiopathogenesis of AA, we conducted a literature search on PubMed and ClinicalTrials.gov. Overall, several authors described interactions between the hair follicles (HFs) and the stress-associated signaling substances, including SP and CRH, in the etiology of AA; this was attributed to the understanding in that AA can occur without the loss of HFs, similar to that observed in hereditary hair loss with age. Most studies demonstrated that the collapse of “immune privilege” plays a crucial role in the development and exacerbation of the AA; nonetheless, a few studies indicated that substances unrelated to autoimmunity may also cause apoptosis in keratocytes, leading to the development of AA. We investigated both the autoimmune and apoptotic pathways within the etiology of AA and assessed the potential interactions between the key substances of both pathways to evaluate potential therapeutic targets for the treatment of AA. Clinical trials of marketed/unreviewed intervention drugs for AA were also reviewed to determine their corresponding target pathways.

## 1. Introduction

Mental stress exhibits various effects on the human body. In particular, it has been revealed that acute stress often elicits immune hyperactivation in the central and peripheral nervous systems, resulting in the development of various autoimmune diseases [1,2]. Alopecia areata (AA) is one of these autoimmune diseases, and is considered as a common, chronic, inflammatory dermatological condition. The worldwide prevalence of AA is approximately 2% within the general population, with no significant differences reported between sex, age, or ethnicity [3,4].

Overall, heredity, psychological stress, and lifestyle are recognized to be the main causes of AA. People with specific hereditary characteristics possess a higher probability of developing AA compared to the general population; however, with an estimated occurrence of 10–20% within such hereditary groups, AA appears to possess a multifactorial genetic pattern that is not in accordance with the simple Mendelian genetics [5]. In a prior study, among the 23% of participants who experienced AA, this disease was attributable to psychological stress in almost twice as many patients than in those without stress [6]. Additionally, patients with certain autoimmune diseases, such as allergies, psoriasis, thyroid disease, and vitiligo, are more likely to develop AA; however, the exact cause for this increased susceptibility remains unknown [7,8]. Normal immune cells do not harm the body nor the hair follicles (HFs). However, it has been hypothesized that when the immune system changes due to various reasons, immune cells can cause the inflammation of the HFs, resulting in the development of AA [7,8]. As AA disrupts hair growth patterns via HF inflammation, it is therefore necessary to differentiate between the changes that arise as part of the normal hair growth cycle, and changes that occur due to the development of AA [9].

The HF is a skin organ that regulates hair growth in mammals and is affected by various factors, such as immune cells, neuropeptides, and hormones [10,11,12]. The structures within the lower part of the HF that are associated with AA include the dermal papilla (DP), inner root sheath (IRS), outer root sheath (ORS), and connective tissue sheath (CTS). The DP is a region in which epithelial cells actively divide to form hair. The IRS directly affects hair formation via the DP and the cell layer surrounding the hair. In addition, the IRS is surrounded by an ORS. The CTS is a thin-membrane tissue surrounded by muscle fibers that are entangled to provide a supportive structure for other tissues. Within the HFs, a CTS surrounds the ORS [13].

AA is a disease that is associated with the HFs. There are various causes of AA development, including overexertion, bacterial infection, and damage to the HFs. AA is also a key inflammatory skin disease that occurs locally on the scalp; specifically, AA is an autoimmune disease caused by CD8+ T cells attacking the HF due to abnormal endocrine responses [14]. Although AA is not a life-threatening nor a physically painful disease, many patients experience psychological pain.

Based on these findings, research has been conducted on the psychopathology of AA. Typically, during hair growth, hair is formed through a cycle of growing, catagen, resting, and shedding. During this process, immune cell infiltration does not occur in the HFs; however, immune cell dysregulation can occur in the HFs during AA [15]. This suggests that nerve-mediated inflammation in the HF disrupts the immune privilege (IP) of the follicular epithelial cells, resulting in the development of autoreactive CD8+ T cells that target the HFs and lead to the development of AA [16,17]. 

AA onset is associated with substance P (SP) and corticotropin-releasing hormone (CRH). The correlation between these two substances and AA is consistent with the hypothesis in that AA is an autoimmune disease [18,19,20]. However, the mechanism underlying AA development has yet to be fully elucidated; therefore, there may be additional unknown pathways involved in this mechanism. For example, SP and CRH may be involved in the development and exacerbation of AA through apoptotic pathways, alongside their roles within the autoimmune pathway.

Therefore, the aim of this review was to summarize the studies that evaluated the process of AA development and its association with mental stress. Specifically, this review focused on two substances, SP and CRH, whose roles have been divided into two aspects: autoimmune and apoptotic pathways.

### 1.1. Role of Substance P and Corticotropin-Releasing Hormone in the Alopecia Areata

Neuropeptides, which are secreted in response to psychological stress, such as SP and CRH, participate in the induction of AA [21,22]. SP is a tachykinin neuropeptide composed of 11 amino acids [23]. As a neurotransmitter and neuromodulator that delivers inflammatory and pain signals, SP initiates the nerve response to stress. Additionally, it is involved in initiating immune-stimulating signals and in controlling inflammatory reactions, cell proliferation, and apoptosis [19].

Although it remains unclear as to how psychological stress stimulates the secretion of stress mediators by the central nervous system, it has been confirmed from previous studies in that this stimulus upregulates the level of SP in the peripheral nervous system of various organs, including the human scalp [19,20]. Mice exposed to psychological stress via short-term restraints (holding the tail for 1 min) exhibited increased levels of SP-like immunoreactivity in the periaqueductal gray region of the brain [24]; further, mice exposed to immobilization stress for 20 min exhibited a pronounced increase in SP release within the amygdala [25]. Upregulated SP release induces HF inflammation and destruction of the intermediate filament of the HF; additionally, SP secreted from the cutaneous sensory nerve ending has been observed to inhibit hair growth and cause neurogenic dermatitis, which induces premature catagen [26]. Recent studies have demonstrated that serum SP levels are increased in patients with AA. In addition, immunohistochemical analysis revealed that the expression of SP, the SP receptor, and the neurokinin-1 receptor (NK1R) was higher in the scalps of patients with AA [26]. In C3H/HeJ mice at the initial stages of AA development, the number of intra/subepidermal and dermal SP-immunoreactive nerve fibers was determined to be significantly higher than that in non-affected mice. Additionally, SP treatment results in an increase in granzyme B expression in CD8+ cells and may therefore lead to a corresponding increase in the levels of cytotoxic activity present in the skin affected by AA, especially within the HF epithelium [27].

CRH is a neuropeptide hormone that is secreted by the hypothalamus and is released in response to unexpected psychological stress [18]. Its primary function is to promote the synthesis and secretion of adrenocorticotropic hormones (ACTHs) through the pituitary gland. ACTH promotes the synthesis of glucocorticoids, such as cortisol, by the adrenal cortex; these glucocorticoids exhibit extensive effects on immunometabolic homeostasis throughout the body [28]. A significant increase in the number of CRH-immunoreactive cells in the paraventricular nucleus of the hypothalamus (PVN) has been observed in rats subjected to restraint stress for 2 h daily for 15 days; further, this CRH upregulation persisted for a considerable period, even after 25 days without this stress [29]. In addition, the expression of the CRH, CRHR1, and c-Fos genes was observed to be higher in the PVN, hippocampus, and amygdala of rats subjected to chronic mild stress. This increase in CRH has been understood to dysregulate the hypothalamic–pituitary–adrenal (HPA) axis and inflammatory responses in the brain [30]. In addition to relaying stress signals through the HPA axis, CRH promotes the maturation of mast cells and the secretion of granules in peripheral tissues, further stimulating the development of AA [31]. By observing CRH receptor expression in the skin of patients with AA after acute emotional stress, it was demonstrated that CRH type 2 beta receptor expression is significantly increased in the skin surrounding the HFs in those with AA [32]. Overall, it was shown that CRH, ACTH, and α-melanocyte-stimulating hormone (α-MSH) expression were all significantly higher in the scalp epidermis and HFs of patients with AA [33]. These results indicate that a defective signaling system in the cutaneous HPA axis is involved in the pathogenesis of AA. 

Therefore, it was confirmed that stress initiators, such as SP and CRH, cause the development of AA [34]. CRH and SP are secreted by mast cells in the dura and colon of rats subjected to acute psychological stress following restraint stress for 30 min; the release of CRH and SP then induces mast cell degranulation [35]. Expression of SP and corticotropin-releasing hormone receptor-1 (CRHR-1) has been observed to be significantly increased in the lesioned skin of patients with psoriasis, suggesting that SP can stimulate the mast cells and increase the expression of functional CRHR-1; further, SP was shown to induce neurokinin-1 (NK1R) gene expression [36].

### 1.2. Neurokinin-1 Receptor Expression in the Human Hair Follicle 

NK1R is the receptor with the highest affinity to SP. NK1Rs are widely distributed throughout the human body. Specifically, it is present in the central and peripheral nervous systems, blood vessels, muscles, lungs, thyroid, and various immune cells [26]. The SP signal is secreted at the end of the peripheral nerve; then, when SP binds to NK1R, it amplifies signaling through secondary transporters, such as cyclic AMP (cAMP), to regulate the selective transcription of each tissue cell [37]. 

The HF is a pocket-shaped organ in the dermal layer of the skin that is composed of 20 different cells. HFs regulate hair growth through complex interactions between the neuropeptides, hormones, and their closely associated immune cells [38]. When HFs are in their anagen phase, an actively growing phase of the human scalp, NK1R expression remains stable [39]. The region of the HF in which NK1R is prominently expressed is dependent on whether the HF is located in the proximal or distal region of the human scalp. The distal part of the human scalp contains HFs that highly express NK1R in the ORS, whereas the proximal part possesses HFs that highly express NK1R in the IRS. The thicker ORS and thinner IRS wrap around the HF; this provides a supply of stem cells to form new hairs and functions as a supportive structure that helps to shape and grow hairs, alongside many other roles [40]. All cells that express NK1R, with the exception of keratinocytes in the ORS of follicles in the distal part of the human scalp, are characterized with slow proliferation and complete keratinization due to terminal differentiation. When terminal differentiation occurs, the cells can no longer divide nor proliferate. These cells can only differentiate to form a granule layer, and eventually develop into a stratum corneum that is composed of dead cells [26]. 

Therefore, the observation in that hair within a specific region of the scalp falls out first in AA can be explained. Hair from HFs in the proximal part of the human scalp fall out first as NK1R is distributed in the IRS. The hair from HFs in the distal part of the human scalp fall out later as NK1R is distributed in the ORS. Thus, the distribution of NK1Rs in HFs is an important factor for AA as the terminal differentiation of the IRS influences hair shaft growth more significantly than that of the ORS [26]. 

### 1.3. Role of CD8+ T Cells in Alopecia Areata 

A genome-wide study identified 14 loci that were significantly correlated with proto-apoptosis [41]. Many of these proteins are involved in regulating immune responses and belong to the human leukocyte antigen (HLA) domain. The HLA region is a DNA sequence located on chromosome 6, which encodes the amino acid sequences of proteins that play a key role in immunity. This sequence is closely related to various autoimmune diseases (including rheumatoid arthritis and type 1 diabetes), as it encodes the protein required for CD4+ T cell antigen presentation following the uptake of the corresponding antigens [42].

Torales et al. argued that the expression of genes encoding CD8, NKG2D, ULBP-3, and ULBP-6 plays a significant role in AA development [43]. Notably, these membrane proteins are all expressed by CD8+ T cells, and their expression is controlled by stress hormones.

### 1.4. Apoptotic Pathway in Alopecia Areata Pathology

SP and CRH are two signaling molecules that serve as key initiators of AA development. Substances related to the apoptotic pathway are indicated in green in Figure 1. SP downregulates immunoreactivity to tropomyosin receptor kinase A (TrkA) and upregulates the p75 neurotrophin receptor (p75NTR), thereby inducing apoptosis [44,45]. Alternatively, CRH directly promotes apoptosis through the tumor necrosis factor (TNF) signaling pathway [46]. Furthermore, nerve growth factor (NGF) is synthesized in the HF when the concentration of SP is increased by the nearby peripheral nerves. NGF, a neurotrophic factor that controls cell proliferation and apoptosis, is secreted from the HFs and binds to the cell surface receptors, such as TrkA and p75NTR, within this follicle [47]. Upon binding, TrkA and p75NTR transmit intracellular signals that determine whether to promote cell proliferation and apoptosis, respectively [26]. TrkA, the primary receptor for NGF, mediates survival, neurite outgrowth, synapse strengthening, and differentiation when bound to NGF. Pro-survival signaling pathways that are triggered by TrkA include the Ras-mitogen-activated protein (MAP) kinase, phosphatidylinositol 3-kinase (PI3K)/protein kinase B (PKB or AKT), and phospholipase C-gamma (PLC-γ) pathways [48]. Conversely, p75NTR signaling initiate apoptosis, the retraction/inhibition of neurite outgrowth, and synapse weakening, following binding to its corresponding ligand, NGF. Apoptosis is activated by various p75NTR signaling pathways, including sphingomyelinase/ceramide, caspases 3, 6, and 9, c-Jun phosphorylation, and the c-Jun N-terminal kinase (JNK) pathways. When little-to-no TrkA is expressed in cell culture systems, p75NTR typically mediates apoptosis. These processes ultimately culminate in the development of AA [49].

Peters et al. cultured human scalp skin HFs in the anagen phase and treated them with SP for 3 days [26]. In this study, the expression of p75NTR was determined to be upregulated in the proximal ORS and CTS. Additionally, the intrafollicular levels of NGF were also upregulated [26]. In contrast, TrkA expression was negligible within the ORS of HFs treated with 10^−8^ and 10^−10^ M SP, and remarkably decreased with 10^−12^ M SP treatment, respectively [26]. These findings suggest that SP promotes apoptosis in HF cells by upregulating the intrafollicular levels of NGF and p75NTR and downregulating TrkA protein expression.

Keratinocytes, located in the bulge of the lower ORS in HF, undergo apoptosis upon the degranulation of mast cells in response to CRH, thereby releasing the bioactive substance tumor necrosis factor-α (TNF-α) [50]. Upon binding to TNF-α, the TNF-α receptors (TNFRs) form complexes that interact with their signaling mediators, leading to the activation of signal transduction pathways and the subsequent induction of apoptosis in HFs. This signaling cascade ultimately results in AA [51].

Following TNF-α–TNFR binding, homotrimerization of the TNFR subunit is initiated, resulting in the formation of a death domain structure, which is necessary for their corresponding interaction with the death domain of the TNFR-associated death domain protein (TRADD). This binding of the death domains is followed by the recruitment of several signaling molecules, such as the Fas-associated protein with death domain (FADD), receptor-interacting serine/threonine-protein kinase 1 (RIPK1), and TNFR-associated factor (TRAF), to this receptor complex. Subsequently, several downstream signaling pathways, including the apoptosis and nuclear factor kappa B (NF-κB) signaling pathways, are activated [52].

The primary modulation of the TNF-α-induced apoptosis is mediated by TNFR1 signaling. To activate this pathway, the ‘death-inducing signaling complex (DISC)’ is formed by TRAF1, FADD, and TRADD; additionally, the silencer of the death domain (SODD) is released to form the DISC. The formation of this complex is followed by the recruitment and activation of the initiator caspases, such as caspase 8, resulting in the activation of executioner caspases, such as caspases 3 and 7. Among these executioner caspases, TNF-α-induced apoptosis is primarily mediated by caspase 3, which can activate a DNase, leading to the degradation of genomic DNA. Therefore, caspase 3 is directly linked to cell death, and TNF-α can strongly induce apoptotic cell death [53].

Kasumagic-Halilovic et al. evaluated serum TNF-α levels in control subjects and in patients with AA using an enzyme-linked immunosorbent assay technique. The mean serum TNF-α was found to be significantly higher in patients with AA than that in the control group of participants (10.31 ± 1.20 vs. 9.59 ± 0.75 pg/mL [mean ± SD], respectively; *p* = 0.044) [54]. Additionally, Lis et al. observed a significant elevation in the serum levels of TNF-α receptor type I in patients with AA when compared to their control subjects [55]. These findings suggest that TNF-α, a signaling molecule derived from mast cells, may be positively correlated with the pathogenesis of AA via the induction of cell apoptosis [55].

Although the apoptotic pathway has a marked influence on the development of AA, its effects are not exclusive to the autoimmune pathway. Therefore, we suggest that the apoptotic pathway is not the primary pathway implicated in AA development. 

### 1.5. Autoimmune Pathway in Alopecia Areata Pathology

The peripheral nerve terminals of the sensory neurons and the epidermal and dermal layers of the skin interact closely, with both organs secreting compounds that support the regulation of the other. It has been suggested that SP and CRH are secreted by sensory nerves in the dermis when an individual is under psychological stress; additionally, these neuropeptides are associated with the development of various autoimmune diseases [56]. This understanding has led to the hypothesis in that SP and CRH initiate an autoimmune pathway in the development of AA.

Substances associated with autoimmune pathways are indicated in blue in Figure 1. SP and CRH stimulate mast cell degranulation, resulting in neurogenic inflammation near the HFs. Additionally, CRH has been previously demonstrated to stimulate plasmacytoid dendritic cells, which can segregate interferon (IFN)-α [56]. In addition, when CRH activates CD8+ T cells by inducing the release of neurogenic inflammatory cytokines via the degranulation of the mast cells, CD8+ T cells can lead to a shift from an immunosuppressive to inflammatory phenotype, as they self-activate via positive feedback between interleukin (IL)-15 and IFN-γ. Ultimately, this can lead to the collapse of the IP of the HFs, which is a critical point in the development of AA.

Overall, CRH and SP promote the degranulation of the mast cells near the HFs. According to a study, in which human scalp skin HFs in the anagen phage were cultured for 3 days with or without SP treatment, control HFs possessed a lower proportion of degranulated mast cells in the CTS compared to the HFs treated with 10^−10^ M SP, which resulted in a 100% degranulation of the mast cells [26,57].

The degranulation of mast cells induces neurogenic inflammation via various signaling molecules, such as histamine [56]. Neurogenic inflammation alters hair growth by promoting the progression of HF into the catagen phase at an earlier time point than that for normal anagen HFs [26,58]. Additionally, SP was found to promote the development of a catagenic morphology in organ-cultured human anagen HFs [59]. Following treatment of dermal papillae with SP, the area of proliferating keratinocytes containing the antigen Ki-67, a cellular marker for proliferation, has been observed to decrease; additionally, terminal deoxynucleotidyl transferase dUTP nick end labeling (TUNNEL)-positive apoptotic cell nuclei have been observed following this treatment [26,31,34]. 

The regression stage of the hair cycle, when HFs forcibly enter into the catagen phase, is the critical point of AA development and indicates the collapse of IP. HFs achieve IP through both active and passive strategies. A key passive strategy in this process is the downregulation of β2 microglobulin, a component of the heterodimeric major histocompatibility complex (MHC) class I; this then results in the inhibition of the ectopic expression of the MHC class I molecules. Downregulation of MHC class I expression in anagen HFs prevents the stimulation of immune cells, such as natural killer and cytotoxic T (CD8+ T) cells [3]. The active strategy for MHC downregulation includes the local secretion of immunosuppressants and the expression of immunoinhibitory signaling molecules. For example, CD200 is a “no danger” signal for HFs, which protects epidermal cells from autoimmune destruction [60]. Recently, Janus kinase (JAK)—signal transducer and activator of transcription (STAT) inhibitors have emerged as promising therapies for AA. These JAK-STAT pathways are intracellular signaling pathways that transmit IL and IFN signals from the cell membrane to the nucleus. Extracellular ligands bind to the type 1 or 2 receptors that activate JAK proteins intracellularly and phosphorylate heterotrimeric STAT proteins, which, in turn, translocate to the nucleus to regulate gene expression. Clinical trials of tofacitinib, the first tested and approved oral JAK (primarily JAK1 and JAK3) inhibitor in humans, have demonstrated that this inhibitor is an effective treatment for autoimmune diseases, including psoriasis and psoriatic arthritis. Upadacitinib is also known to be effective in treating rheumatoid arthritis, psoriatic arthritis, and atopic dermatitis by inhibiting the actions of the JAK-STAT pathways as a target for inflammatory responses [61]. Additionally, treatment with tofacitinib (15 mg/day) for >5 months exhibited a therapeutic effect against AA (Table 1) [62]. Moreover, as shown in Table 1, the AA-targeted drugs in phase 2 and 3 clinical trials include ruxolitinib (a drug ointment) and ritlecitinib (oral medication, tablet), both of which can target JAK-STAT inhibition [63,64]. Extending from this, phosphodiesterase (PDE)-4 inhibitors (apremilast and secukinumab) and pro-inflammatory cytokine inhibitors, such as IL-4 and IL-13 (dupilumab) have a limited efficacy. For example, although not yet clinically approved, risankizumab is also known to be effective in treating psoriasis-type alopecia by suppressing the increase in pro-inflammatory cytokines, such as TNF-α, IL-12, IL-13, and IL-17 [65]. Nonetheless, ongoing long-term studies and future long-term investigational therapies may be able to further demonstrate the usefulness of these novel therapies for the treatment of AA (Table 1) [66]. 

No inflammatory cells are present around normal HFs in the late anagen phase in individuals with AA, indicating an aberrant expression of the MHC class I and II molecules [60]. The CTS of SP-treated dermal papillae exhibited an upregulation of MHC class I expression; additionally, keratocytes in the HF matrix also expressed ectopic MHC class I molecules. In some cases, the IRS and ORS of the HFs also expressed ectopic MHC class I, and this upregulation was found to be proportional to the SP concentration [67]. It has been understood in that the expression level of β2 microglobulin is similar to that of MHC class I in the HFs. As observed in MHC class I, the expression of β2 microglobulin in the HF matrix, keratinocytes, CTS, and dermal papilla in the control group was found to be very low or absent, whereas the corresponding expressions in the SP-treated group were significantly higher [26]. This may indicate that the downregulation of MHC class I in the normal anagen HFs is achieved via the suppression of β2 microglobulin expression, since β2 microglobulin is an essential part of the MHC class I molecule.

However, CRH stimulates plasmacytoid dendritic cells (pDCs) which secrete IFN-α [68]. Thus, CD8+ T cells are subsequently activated by IFN-α secreted from the pDCs, leading to further secretions of IFN-γ by the CD8+ T cells. IFN-γ secreted by the CD8+ T cells promotes the secretion of IL-15 in the HFs. IL-15 also promotes the secretion of IFN-γ in the CD8+ T cells, which forms a positive feedback loop. CD8+ T cells expand and are activated by multiplying and polarizing the clonal cells in this cycle, which results in an increase in the number of CD8+ T cells that are able to target the HFs. 

Through this process, CD8+ T cells can shift from an immunosuppressive to inflammatory phenotype upon their contact with mast cells [69]. This is accomplished by the increased expression of membrane proteins, such as the 4-1BB ligand, OX40 ligand (also known as CD134 or TNFRSF4), intercellular adhesion molecule-1 (ICAM-1), and the CD30 ligand on the mast cells. The suppressed expression of programmed cell death ligand 1 (PD-L1) and the increased tryptase secretion by the mast cells are also involved in this process. Inflammatory T cells are more prone to attack HFs than the immunosuppressive type, thus increasing the risk of autoimmune diseases. 

Overall, it can be inferred that CD8+ T cells in patients with AA can bind to the MHC class I molecules without antigens, as mast cells already express these specific membrane proteins. When ICAM-1 or 4-1BB ligands bind to membrane receptors, such as lymphocyte function-associated antigen-1 (LFA-1), in CD8+ T cells, mast and CD8+ T cells come into long-term contact. During this extended period of contact, there is an increased probability of MHC class I molecules binding to the T cell receptors on CD8+ T cells without antigens.

When CD8+ T cells are activated by mast cells, they secrete IFNs, which not only disrupt the IPs of HFs, but also allow the HFs to enter the catagen phase early [70]. A prior study reported that signal-transmitting factors that participate in the apoptotic pathway could also be involved in the autoimmune pathway. The binding of NGF to TrkA promotes HF transition into the anagen phase; further, the interaction between proNGF/p75NTR is correlated with the stimulation of early entry into the catagen phase [71]. Adly et al. determined that Trk A and p75NTR protein expression levels significantly fluctuate depending on the hair cycle stage during the transition from the anagen to the catagen to finally the telogen stages. In response to high NGF expression levels, Trk A, the corresponding receptor with a high affinity to NGF, was also strongly expressed in human scalp HFs in the anagen phase; however, this receptor was either absent or weakly expressed in the catagen and telogen phases of the HFs [72]. In contrast, after a few days of organ culture, catagen and telogen HFs exhibited significantly higher expression levels of p75NTR compared to the HFs in the late anagen phase. p75NTR moderates the anagen–catagen transition of HFs during the late anagen phase. In addition, p75NTR knockout animals exhibited a greater delay in the catagen phase transition compared to their wild-type controls [72]. Notably, substances related to apoptotic pathways can also induce the HFs to enter the catagen phase earlier than normal HFs. This indicates that the two pathways may closely interact with each other. In addition, p75NTR knockout animals demonstrated a much greater catagen delay than the wild-type controls [72].

Notably, substances related to apoptotic pathways have also been shown to possess the ability to induce an earlier transition of the HFs into the catagen phase. This indicates that the autoimmune and apoptotic pathways may interact with each other. 

### 1.6. Improved Therapeutic Strategies Targeting Signaling Pathways for Alopecia Areata 

It is known that various growth factors and signaling pathways are involved in the hair development process. Although diverse intercellular signaling pathways are intricately involved in hair development, the Wnt/β-catenin signaling pathway plays a key role in stimulating hair follicle stem cells and hair regeneration [73,74]. A few studies have suggested that various Wnts promote hair cycling and regeneration via the activation of β-catenin signaling. For example, intradermal injections of Wnt1α-enriched conditioned media from bone marrow mesenchymal stem cells into depilated mouse skin accelerated HF progression from telogen to anagen, increased the amount of hair, and elevated the expression of hair induction-related genes, such as Lef1, versican, and Gli1 [75]. However, the topical application of tocotrienol-rich formulation (TRF, 5 mg/cm^2^) on the depilated dorsal skin of healthy or diabetic mice markedly induced epidermal hair follicle development and early anagen induction, as evidenced by change in skin color from pink to black [76]. 

It is also known that apoptotic macrophages can activate epithelial hair follicle stem cells (HFSCs) in a Wnt-dependent manner, and the inhibition of macrophage-derived Wnts can delay the anagen phase. Thus, beyond their function as phagocytes, macrophages contribute to the cyclical activation of HFSCs as regulators of tissue regeneration and organ remodeling [77]. 

Additionally, mitochondrial pyruvate carrier (MPC) inhibitors are known to increase lactate dehydrogenase (LDH) activity in HFSCs. When MPCs are inhibited, cytoplasmic pyruvic acid cannot enter mitochondria, but instead is converted into other metabolites, such as lactate by LDH, increasing LDH activity and activating HFSCs. Accordingly, MPCs promote hair growth [78]. 

Autophagy is a cellular disposal system which removes unfolded proteins and damaged organelles and provides nutrition through lysosomal degradation and recycling of breakdown products. It has been known to play an important role in response to stress in recent years. It revealed that variations in genes, such as *STX17*, *CLEC16A*, and *BCL2L11*/*BIM*, which all play a role in regulating autophagy, are predisposing genetic loci for AA [79]. For example, AA mice treated with the autophagy-inducing Tat-BECN1 peptide displayed attenuated hair loss, whereas blocking autophagy with chloroquine resulted in accelerated hair loss [79]. Several studies have revealed that in anagen, hair matrix keratinocytes (MKs) of organ-cultured HFs exhibit an active autophagic flux, as previously documented by the evaluation of the endogenous lipidated light chain 3B (LC3B) and sequestosome 1 (SQSTM1/p62) proteins, and the ultrastructural visualization of autophagosomes at all stages of the autophagy process [80]. Therefore, autophagy can act as a promising drug for the treatment of hair growth disorders and drug-induced alopecia characterized by an early catagen induction. 

### 1.7. Potential Therapeutic Treatments for Alopecia Areata

The IP of HFs is first disrupted by the ectopic expression of MHC class I molecules. In addition, when CD8+ T cells are exposed to ectopic MHC class I molecules, the expression levels of the MHC class II molecules increase, resulting in secondary autoimmune interactions. The primary autoimmune effects can be inhibited by α-MSH, transforming growth factor-β1 (TGF-β1), and insulin-like growth factor-1 (IGF-1); meanwhile, secondary autoimmunity is inhibited by α-MSH, TGF-β1, and IL-10. These inhibitory mechanisms function to restore the IP of the HF when psychological stress ceases. α-MSH and TGF-β1 are strong natural immunosuppressants and serve to suppress the first and secondary autoimmune phenomena. These factors also promote the maintenance of the anagen phase for a longer period, inhibit entry into the catagen phase, and suppress the ectopic expression of MHC class I in the HFs. As α-MSH and TGF-β1 treatments have a relatively low potential risk of triggering side-effects, a promising therapeutic strategy could be the localization of α-MSH and TGF-β1 upregulation to certain HFs via follicular-targeting liposomal drugs. 

As a direct method of treatment, the use of drugs can alleviate AA through IP regeneration in the HFs. Calcineurin inhibitors, such as FK506 (tacrolimus), downregulate MHC class I expression. These inhibitors have also been shown to contribute to the prevention of IP collapse induced by IFN-γ in ex vivo studies [58]. However, the effective delivery of these drugs to the corresponding HFs remains unclear.

As discussed earlier, failure of the IP mechanism triggers AA development. Therefore, normalizing IP may be the most effective way to treat AA. This has been previously achieved through AA treatment with ultraviolet A-1 (UVA-1) phototherapy, which causes the amount of α-MSH to increase [58]. Consequently, α-MSH can then help in alleviating AA. In addition, diphencyprone treatment promotes the expression of IL-10 and TGF-β1; this normalizes the expression of MHC molecules in the HF, which is otherwise eliminated in autoimmune diseases, including in AA (Table 1) [58]. This treatment also indirectly aids in the restoration of IP in patients with AA.

As previously explained, there is a therapeutic method for treating AA using UVA-1 that increases α-MSH levels. Therefore, α-MSH analogs, such as afamelanotide and melanotan, could be directly used as therapeutic drugs. These α-MSH analogs act as α-MSH by inhibiting the ectopic expression of the MHC class I molecules, and thus inhibiting their corresponding autoimmune effects [58]. These inhibitors also play a key role in inducing the production of other IP defensive mechanisms [58]. As a result, α-MSH analogs are strong candidates for the treatment of AA.

SP promotes AA development through various mechanisms. Therefore, antagonists of SP receptors, such as aprepitant, netupitant, and rolapitant, may also exhibit positive effects in the treatment of AA via the inhibition of the AA-related signaling mechanism initiated by SP [58]. 

As shown in Table 1, the AA-targeted drugs in phase 4 clinical trials include imiquimod, a drug (ointment) that targets TLR7, and tofacitinib, a drug (oral medication, tablet) that targets JAK-STAT inhibition. Additionally, AA-targeted drugs currently undergoing phase 2 clinical trials include ruxolitinib and ritlecitinib, which target JAK-STAT inhibition, and triamcinolone, a drug (subcutaneous injection) that targets cytokines. Further, many drugs are currently in the clinical stages for the treatment of AA (Table 1). Also, the AA-targeted drugs in phase 2 clinical trials include tralokinumab, a drug (subcutaneous injection) which targets IL-13 inhibitors, triamcinolon, a drug (through oral medication, subcutaneous/muscle injection, inhalation administration methods) which targets glucocorticoid receptor agonists, and naltrexone, a drug (oral medication and intramuscular injection) which targets μ-opioid receptor antagonists (Table 1). Ingenol, mebutate, and LEO43204, which are being evaluated as AA treatments in phase 2 and 3 clinical trials, have not yet been accurately targeted. Also, although not yet clinically approved, alefacept is also being used as a treatment for AA as a CD2-targeting inhibitor.

## 2. Conclusions

Overall, we reviewed several studies that have examined how psychological stress causes AA. As AA is an acquired autoimmune disease, the autoimmune pathway plays a critical role in the development of this disease. We noted that only tissues with HFs expressing high levels of NK1R, an SP receptor, are characterized with slow proliferation and active complete hallucinogenic keratinization, which has been attributed to terminal differentiation; additionally, several studies have reported that SP and CRH, initiators of AA, not only disrupt the IP of HFs, but also promote the apoptosis of the HF cells. Therefore, we concluded that the mechanism of stress-induced AA involves two pathways: autoimmune and apoptotic pathways. The autoimmune pathway involves the ectopic expression of MHC class I in the HFs following neurogenic inflammation initiated by various signals, resulting in the collapse of IP and causing CD8+ T cells to attach to HF keratocytes. The apoptotic pathway regulates various messengers that promote apoptosis in HF keratocytes. Although the autoimmune pathway plays the most decisive role in the pathogenesis and development of AA, we expect that the apoptotic pathway also affects the progression of AA to some extent. For example, apremilast reversibly inhibits the autoimmune pathways by selectively inhibiting phosphodiesterase 4; nonetheless, apremilast also acts as an inhibitor of TNF-α, one of the key messengers in the apoptotic pathway. Many studies have been conducted to uncover the mechanisms underlying AA development; however, a complete understanding of this mechanism has not yet been elucidated. To develop innovative treatments for AA, the mechanism of etiology needs to be fully understood; therefore, further research is required to explore this mechanism in depth.

## Figures and Tables

**Figure 1 ijms-24-11711-f001:**
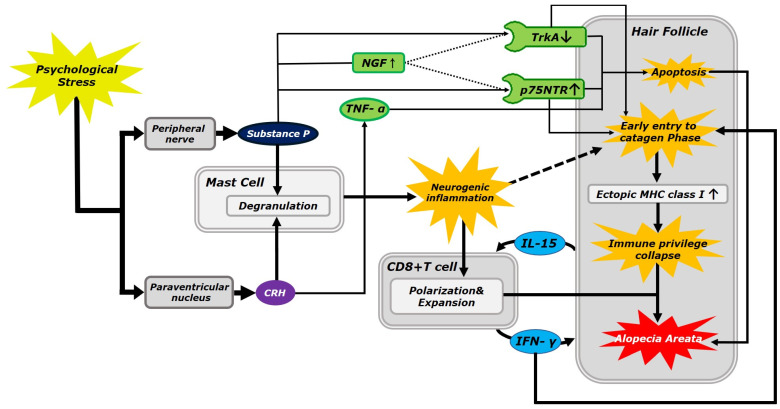
A schematic detailing the psychopathology of alopecia areata. Autoimmune and apoptotic pathways are detailed; however, only approximate flows are shown, and detailed signaling processes are excluded for conciseness. The autoimmune pathway, the major pathway in alopecia areata development, is indicated by a thick line; ligands associated with the autoimmune pathway are shown in blue. The reason for which neurogenic inflammation stimulates hair follicles to enter the catagen phase early is unclear; therefore, this is indicated by a dotted line. The apoptotic pathway, the minor pathway in alopecia areata development, is indicated by a thin line; ligands associated with the apoptotic pathway are shown in green. The apoptotic pathway interacts with the autoimmune pathway by downregulating the expression of tropomyosin receptor kinase A and upregulating the expression of the p75 neurotrophin receptor, which also contributes to the early entry into the catagen phase; therefore, this is indicated by a line with an intermediate thickness. Abbreviations: CRH, corticotropin-releasing hormone; NGF, nerve growth factor; TNF-α, tumor necrosis factor-α; TrkA, tropomyosin receptor kinase A; p75NTR, p75 neurotrophin receptor; CD8+ T cell, cluster of differentiation 8+ T cell; IL-15, interleukin 15; IFN-γ, interferon-gamma; and MHC, major histocompatibility complex.

**Table 1 ijms-24-11711-t001:** Marketed and unreviewed AA treatments (completed clinical studies with results).

Current Status of Clinical Trial	Intervention: Drug Only	Study Title	Drug Target	Drug Application
Phase 4	Tofacitinib (Xeljanz^®^)	Effectiveness and safety of Tofacitinib in patients with extensive and recalcitrant alopecia areata	JAK-1 and JAK-3 (JAK-STAT inhibition)	Oral medication, (tablet)
Phase 2	Ruxolitinib (Jakavi^®^)	A study with ruxolitinib phosphate cream applied topically to subjects with alopecia areata (AA)	JAK-1 and JAK-2 (JAK-STAT inhibition)	Ointment
Phase 2 Phase 3	Ritlecitinib (PF-06651600)	PF-06651600 for the treatment of alopecia areata	JAK-3 (JAK-STAT inhibition)	Oral medication, (tablet)
Phase 4	Apremilast (Otezla^®^)	Apremilast in the treatment of central centrifugal cicatricial alopecia (CCCA)	PDE-4 inhibitor, TNF-α inhibitor	Oral medication, (tablet)
Phase 2	Secukinumab	A study of secukinumab for the Treatment of alopecia Areata	PDE-4 inhibitor	Patches
Phase 2	Dupilumab (Dupixent^®^)	Treatment of alopecia areata (AA) with dupilumab in patients with and without atopic dermatitis (AD)	IL-4 and IL-13 inhibitor	Subcutaneous injection
Phase 3	Diphencyprone	DPCP for the Treatment of Alopecia Areata	IL-10 and TGF-β1 inhibitor	Ointment
Phase 4	Imiquimod (Zyclara^®^)	Characteristics of T cells from alopecia areata scalp skin before and after treatment with aldara 5%	TLR-7 activation	Ointment
Phase 2	Tralokinumab (Adtralza^®^, Adbry^®^)	A pilot study of tralokinumab in subjects with moderate to severe alopecia areata	IL-13 inhibitor	Subcutaneous injection
Phase 2	Triamcinolone (Kenalog^®^)	Adrenal function and use of intralesional triamcinolone acetonide 10 mg/mL (Kenalog-10) in patients with alopecia areata	Glucocorticoid receptor agonist	Oral medication, subcutaneous/muscle injection, and inhalation
Phase 2	Naltrexone (Revia^®^)	Oral low-dose naltrexone for lichen planopilaris and frontal fibrosing alopecia	μ-opioid receptor antagonist	Oral medication (tablet), and intramuscular injection
Phase 3	Ingenol	Efficacy and safety of ingenol mebutate gel for actinic keratosis applied on large area on face, scalp or chest	Not fully understood	Ointment
Phase 3	Mebutate	Efficacy and safety of ingenol mebutate gel for actinic keratosis applied on large area on face, scalp or chest	Not fully understood	Ointment
Phase 2	LEO43204	LEO 124249 ointment in the treatment of alopecia areata	Not fully understood	Ointment
Not applicable	Alefacept (Amevive^®^)	Alefacept in patients with severe scalp alopecia areata	CD2 inhibition	Intramuscular injection

## Data Availability

Not applicable.

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
