# Peer review of "Psychological Stress-Induced Pathogenesis of Alopecia Areata: Autoimmune and Apoptotic Pathways"

_ijms, 2023, doi:10.3390/ijms241411711_

Round 1

Reviewer 1 Report

Dear authors, it is a great idea to present the review concerning molecular mechanisms of alopecia areata and possible treatment of the condition. However my impression from the manuscript submitted is rather negative. The manuscript is replete with lengthy descriptions of commonly known things like hormone cascades, mast cells activations, TNF apoptosis pathway and so on. Along with these excesses, the articles devoted apoptosis related alopecia are sparsely represented. Eg, there is nothing about Wnt/beta-catenin apoptic pathway, nor about mitochondrial apoptosis input to the alopecia progress. There are even no reference on autophagy as the key point of alopecia development. I can't even give specific recommendations for improving the manuscript, just a piece of advice for the authors to be concise and more thorough when reviewing the literature. You may include  the articles of Parodi et al., 2018 and Castellena et al.,  2014 (Macrophages Contribute..), but not just those mentioned as an example.

Author Response

Response) I fully agree with the reviewer's comments. According to the reviewer’s comment, we corrected them in the revised manuscript (heighted in blue on line 398 in page 17) as follows:

1.6. Improved therapeutic strategies targeting signaling pathway for alopecia areata 

It is known that various growth factors and signaling pathways are involved in the hair development process. Although diverse intercellular signaling pathways are intricately involved in hair development, the Wnt/β-catenin signaling pathway plays a key role in stimulating hair follicle stem cells and hair regeneration [74,75]. Some studies have suggested that various Wnts promote hair cycling and regeneration via the activation of β-catenin signaling. For example, intradermal injection of Wnt1α-enriched conditioned media from bone marrow mesenchymal stem cells in to depilated mouse skin accelerated HF progression from telogen to anagen, increased the amount of hair, and elevated the expression of hair induction-related genes, such as Lef1, versican, and Gli1 [76]. However, topical application of tocotrienol-rich formulation (TRF, 5 mg/cm2) on depilated dorsal skin of healthy or diabetic mice markedly induced epidermal hair follicle development and early anagen induction, as evidenced by change of skin color from pink to black [77].

It is also known that apoptotic macrophages can activate epithelial hair follicle stem cells (HFSCs) in a Wnt-dependent manner, and inhibition of macrophage-derived Wnts can delay anagen. Thus, beyond their function as phagocytes, macrophages contribute to the cyclical activation of HFSCs as regulators of tissue regeneration and organ remodeling [78].

Additionally, mitochondrial pyruvate carrier (MPC) inhibitors are known to increase lactate dehydrogenase (LDH) activity in HFSCs. When MPC is inhibited, cytoplasmic pyruvic acid cannot enter mitochondria, but instead is converted to other metabolites such as lactate by LDH, increasing LDH activity and activating HFSCs. Accordingly, MPC promotes hair growth [79].

Autophagy is a cellular disposal system which removes unfolded proteins and damaged organelles, and provides nutrition by lysosomal degradation and recycling of breakdown products. It has been known to play an important role in response to stress in recent years. It revealed that variations in genes such as STX17, CLEC16A, and BCL2L11/BIM, which play a role in regulating autophagy, are predisposing genetic loci for AA [80]. For example, AA mice treated with the autophagy-inducing Tat-BECN1 peptide had attenuated hair loss, whereas blocking autophagy with chloroquine was accelerated hair loss [80].  Some studies have been shown that in anagen, hair matrix keratinocytes (MKs) of organ-cultured HFs exhibit an active autophagic flux, as documented by evaluation of endogenous lipidated Light Chain 3B (LC3B) and sequestosome 1 (SQSTM1/p62) proteins and the ultrastructural visualization of autophagosomes at all stages of the autophagy process [81]. Therefore, autophagy can act as a promising drug for the treatment of hair growth disorders and drug-induced alopecia characterized by early catagen induction.”

Reviewer 2 Report

Very interesting and well written article. 

The authors have conducted an interesting review however there are some important shortcomings in the treatments section. 

1) Table 4 is ok but more treatments that have been reported in the literature should be added in the paragraph to make the review more precise and detailed. Here are some examples that the authors can cite in the text commenting on these drugs 

- Napolitano M, Fabbrocini G, Genco L, Martora F, Potestio L, Patruno C. Rapid improvement in pruritus in atopic dermatitis patients treated with upadacitinib: a real-life experience. J Eur Acad Dermatol Venereol. 2022 Sep;36(9):1497-1498. doi: 10.1111/jdv.18137. Epub 2022 Apr 18. PMID: 35398945.

- Martora F, Villani A, Ocampo-Garza SS, Fabbrocini G, Megna M. Alopecia universalis improvement following risankizumab in a psoriasis patient. J Eur Acad Dermatol Venereol. 2022 Jul;36(7):e543-e545. doi: 10.1111/jdv.18017. Epub 2022 Feb 25. PMID: 35181958.

2) There are some typos in the text I recommend a tatal revision of the text.

3) Small errors of the English language

4) Reference format is not that of the journal, revise according to the guidelines 

Minor editing of English language required

Author Response

Reviewer #2: The authors have conducted an interesting review however there are some important shortcomings in the treatments section. 

1) Table 4 is ok but more treatments that have been reported in the literature should be added in the paragraph to make the review more precise and detailed. Here are some examples that the authors can cite in the text commenting on these drugs. 

- Napolitano M, Fabbrocini G, Genco L, Martora F, Potestio L, Patruno C. Rapid improvement in pruritus in atopic dermatitis patients treated with upadacitinib: a real-life experience. J Eur Acad Dermatol Venereol. 2022 Sep;36(9):1497-1498. doi: 10.1111/jdv.18137. Epub 2022 Apr 18. PMID: 35398945.

- Martora F, Villani A, Ocampo-Garza SS, Fabbrocini G, Megna M. Alopecia universalis improvement following risankizumab in a psoriasis patient. J Eur Acad Dermatol Venereol. 2022 Jul;36(7):e543-e545. doi: 10.1111/jdv.18017. Epub 2022 Feb 25. PMID: 35181958.

Response) I fully agree with the reviewer's comments. According to the reviewer’s comment, we corrected them in the revised manuscript (heighted in blue on line 330 in page 14 and line 475 in page 20) and rearranged the table 1 as follows:

“Also, upadacitinib is known to be effective in rheumatoid arthritis, psoriatic arthritis, and atopic dermatitis by inhibiting the action of JAK-STAT pathways as a target for inflammatory response [62]. Additionally, treatment with tofacitinib (15 mg/day) for >5 months exhibited a therapeutic effect against AA (Table 1) [63]. Also, as, shown in Table, the AA-targeted drugs in phase 2 and 3 clinical trials include ruxolitinib, a drug (ointment) and ritlecitinib (oral medication, tablet) that targets JAK-STAT inhibition [64,65]. Additionally, phosphodiesterase (PDE)-4 inhibitors (apremilast and secukinumab) and pro-inflammatory cytokine inhibitors, such as IL-4 and IL-13 (dupilumab) have limited efficacy. For example, although not yet clinically approved, risankizumab is also known to be effective in psoriasis-type alopecia by suppressing the increase of pro-inflammatory cytokine, such as TNF-α, IL-12, IL-13 and IL-17 [66].”

“Also, the AA-targeted drugs in phase 2 clinical trials include tralokinumab, a drug (subcutaneous injection) that target IL-13 inhibitor, triamcinolon, a drug (oral medication, subcutaneous/muscle injection, inhalation) that target glucocorticoid receptor agonist, and naltrexone, a drug (oral medication, intramuscular injection) that target μ-opioid receptor antagonist (Table 1). Ingenol, mebutate, and LEO43204, which are being tested as AA treatments in phase 2 and 3 clinical trials, have not yet been accurately targeted. Also, although not yet clinically approved, alefacept is also being used as a treatment as an AA targeting a CD2 inhibitor”.

2) There are some typos in the text I recommend a tatal revision of the text.

Response) We agree with the reviewer’s point out. Based on the reviewer’s comment, we checked for spelling and grammatical errors in the revised manuscript, and received English corrections from professional English editors by twice. They were accompanied by a certificate as proof of their professional English editing.

https://app.editage.com/orders/completed. 

3) Small errors of the English language.

Response) We agree with the reviewer’s point out. Based on the reviewer’s comment, we checked for spelling and grammatical errors in the revised manuscript, and received English corrections from professional English editors by twice. They were accompanied by a certificate as proof of their professional English editing.

https://app.editage.com/orders/completed. 

4) Reference format is not that of the journal, revise according to the guidelines.

Response) According to the reviewer’s comment, we corrected them in the revised manuscript.

Round 2

Reviewer 1 Report

Dear Authors,

Thank you for the corrections made.